# LLM2CLIP: Powerful Language Model Unlock Richer Visual Representation

## Abstract

CLIP is one of the most important multimodal foundational models today, aligning visual and textual signals into a shared feature space using a simple contrastive learning loss on large-scale image-text pairs. What powers CLIP's capabilities? The rich supervision signals provided by natural language — the carrier of human knowledge — shape a powerful cross-modal representation space. As a result, CLIP supports a variety of tasks, including zero-shot classification, detection, segmentation, and cross-modal retrieval, significantly influencing the entire multimodal domain. However, with the rapid advancements in large language models (LLMs) like GPT-4 and LLaMA, the boundaries of language comprehension and generation are continually being pushed. This raises an intriguing question: *can the capabilities of LLMs be harnessed to further improve multimodal representation learning?* The potential benefits of incorporating LLMs into CLIP are clear. LLMs' strong textual understanding can fundamentally improve CLIP's ability to handle image captions, drastically enhancing its ability to process long and complex texts — a well-known limitation of vanilla CLIP. Moreover, LLMs are trained on a vast corpus of text, possessing open-world knowledge. This allows them to expand on caption information during training, increasing the efficiency of the learning process. However, realizing this potential is challenging. Despite LLMs' powerful internal comprehension, their autoregressive nature hides this capability within the model, leading to output features with poor discriminability. Our experiments show that directly integrating LLMs into CLIP results in catastrophic performance drops. In this paper, we propose *LLM2CLIP*, a novel approach that embraces the power of LLMs to unlock CLIP's potential. By fine-tuning the LLM in the caption space with contrastive learning, we extract its textual capabilities into the output embeddings, significantly improving the output layer's textual discriminability. We then design an efficient training process where the fine-tuned LLM acts as a powerful teacher for CLIP's visual encoder. Thanks to the LLM's presence, we can now incorporate longer and more complex captions without being restricted by vanilla CLIP text encoder's context window and ability limitations. Our experiments demonstrate that this approach brings substantial improvements in cross-modal tasks. Our method directly boosted the performance of the previously SOTA EVA02 model by 16.5% on both long-text and short-text retrieval tasks, transforming a CLIP model trained solely on English data into a state-of-the-art cross-lingual model. Moreover, when integrated into multimodal training with models like Llava 1.5, it consistently outperformed CLIP across nearly all benchmarks, demonstrating comprehensive performance improvements.

## 1 Introduction

CLIP (Radford et al., 2021) is one of the most important multimodal foundational models today. It aligns vision and language signals into a shared feature space by employing a simple contrastive learning loss on large-scale image-text pairs. As a retriever, CLIP supports a wide range of tasks, including zero-shot classification (Qian & Hu, 2024), detection (Lin & Gong, 2023), segmentation (Zhou et al., 2023), and image-text retrieval (Lülf et al., 2024; Koukounas et al., 2024). As a feature extractor, it has become dominant in virtually all cross-modal representation tasks, such as image understanding, video understanding, and text-to-image/video generation. For instance, works like LLaVA (Liu et al., 2023a) and Qwen-VL (Bai et al., 2023) leverage CLIP as a feature extractor to obtain visual features for text models, while models like Stable Diffusion (Rombach et al., 2021) and DALL·E 2 (Ramesh et al., 2022) use CLIP's text encoder to extract textual features for visual models.

What makes CLIP so powerful, particularly as a vision encoder? The core of its strength lies in its unprecedented ability to align visual pretraining with natural language — the carrier of human knowledge. Unlike earlier vision encoders such as VGG and ResNet, which relied on the limited ImageNet dataset and simple image categories with just a few words, CLIP is trained on web-scale data using rich descriptive text. This alignment with language is what sets CLIP apart and unlocks its vast potential. However, since CLIP's introduction, large language models (LLMs) have advanced significantly. Models like GPT-4 (Achiam et al., 2023) and Llama (Dubey et al., 2024) now demonstrate remarkable language capabilities, yet these advancements have not translated to corresponding

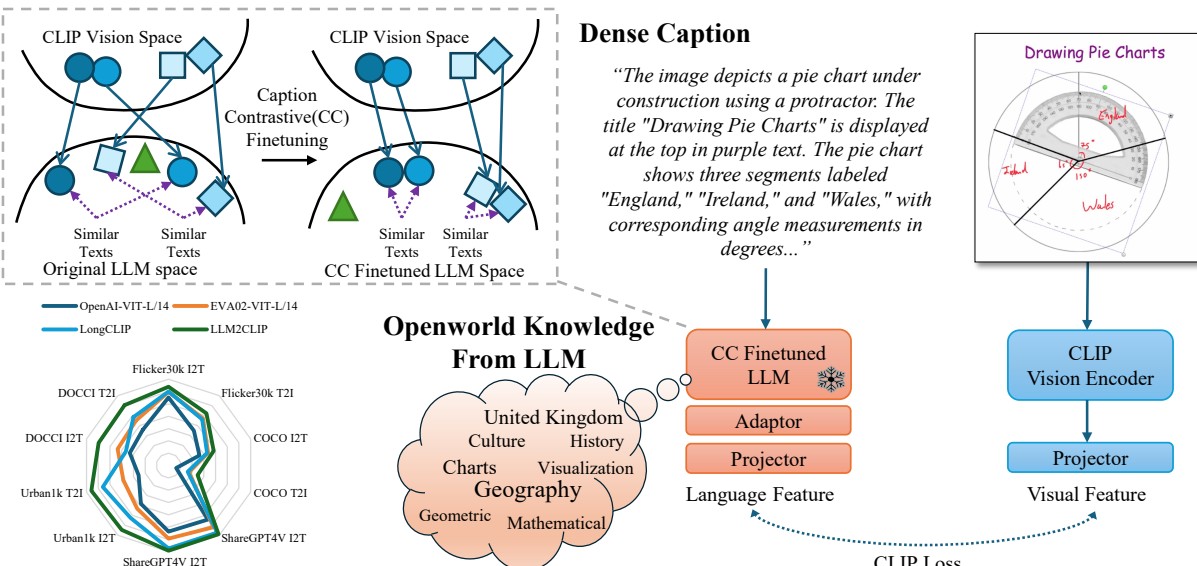

Figure 1: *LLM2CLIP* Overview. After applying caption contrastive fine-tuning to the LLM, the increased textual discriminability enables more effective CLIP training. We leverage the open-world knowledge and general capabilities of the LLM to better process dense captions, addressing the previous limitations of the pretrained CLIP visual encoder and providing richer, higher-dimensional textual supervision. Experimental results demonstrate that LLM2CLIP can make any SOTA CLIP model even more SOTA ever.

improvements in visual representation learning. This prompts the question: *can the capabilities of LLMs be harnessed to further improve multimodal representation learning?*

The potential benefits of incorporating LLMs into CLIP are clear. LLMs' strong textual understanding can fundamentally improve CLIP's ability to handle image captions, drastically enhancing its ability to process long and complex texts — a well-known limitation of vanilla CLIP. Moreover, LLMs are trained on a vast corpus of text, possessing open-world knowledge. This allows them to expand on caption information during training, increasing the efficiency of the learning process.

In this work, we aim to leverage large language models (LLMs) to enable CLIP to learn more powerful, fine-grained, and rich visual representations. Currently, CLIP is often criticized for its bag-of-words-like perception and the limitations of its text encoder, which suffers from a constrained model size, limited context length, and is trained predominantly on image captioning data, lacking exposure to diverse world corpora. A natural approach would be to replace CLIP's text encoder with an LLM that embeds rich human knowledge. However, this presents significant challenges. In the cross-modal contrastive learning framework employed by CLIP, the text encoder functions as a set of knowledge anchors in the shared latent space, guiding the alignment of the vision encoder with human knowledge of the physical world. The structure, richness, and discriminability of these knowledge anchors are critical to the visual model's effectiveness. In contrast, LLMs are primarily designed to predict the next word rather than generate explicit representations of the knowledge they contain. Their textual comprehension abilities and open-world knowledge are latent within the model, rather than present in the output embeddings, making them difficult to utilize in the same explicit manner as CLIP's text encoder. As a result, using LLMs as a text encoder may not produce linearly separable features, which are crucial for effective feature alignment.

To validate our hypothesis, we designed a caption-to-caption retrieval experiment, as shown in Table 1 and Figure 2. Each image in the MS-COCO dataset has five human-annotated captions. We selected the first two captions as positive samples and performed retrieval across the entire validation set. Using the caption retrieval accuracy (CRA), we evaluated the text model's ability to differentiate between captions, helping us determine which language model is better suited for CLIP. We found that Llama-3 8B achieved only 18.4% top-1 accuracy, while the standard CLIP-ViT-L reached 66.0% top-1 accuracy. As illustrated in Figure 2, the top-1 caption retrieved by original Llama-3 can be entirely unrelated to the query caption, clearly obstructing effective CLIP learning. Therefore, directly using an LLM to guide CLIP's visual encoder training is highly constrained.

We believe that enhancing the discriminative power of LLM output tokens through fine-tuning is vital for the success of our proposed approach, allowing the latent capabilities of LLMs to surface. Encouragingly, we found that this can be achieved very efficiently. Specifically, we designed a caption contractive (CC) fine-tuning strategy, applying lightweight fine-tuning to the output tokens of Llama-3 8B using LoRA on the CC3M (Sharma et al., 2018) image captioning dataset. The primary goal of this training task was to adjust the output space, improving the model's ability to distinguish between different captions. We utilized a supervised SimCSE (Gao et al., 2021;

Table 1: **Comparison of top-1 Caption Retrieval Accuracy (CRA) for various language models in MS COCO validation set.**

| Language Model | CRA |
|---|---|
| CLIP-L/14 | 66.6 |
| EVA02-L/14 | 69.8 |
| Llama3-8B | 18.4 |
| Llama3.2-1B | 18.3 |
| Llama3-8B-CC | **73.0** |
| Llama3.2-1B-CC | 72.8 |

| | | |
|---|---|---|
| Query: | *"people on bicycles ride down a busy street"* | |
| Llama3: | *"a woman sits on a brief case in the woods "* | |
| Finetuned: | *"A group of people are riding bikes down the street in a bike lane"* | |
| Query: | *"A small blue plane sitting on top of a field."* | |
| Llama3: | *"A man standing on a field, holding a bat."* | |
| Finetuned: | *"an E2 airplane painted blue with black and white stripes "* | |

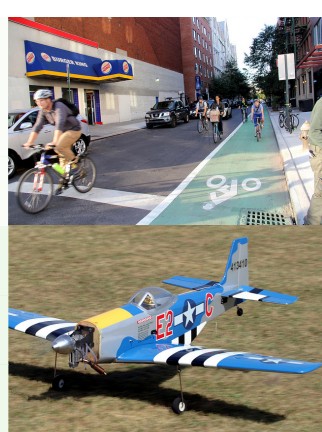

Figure 2: Real examples of top-1 results from the caption-to-caption retrieval experiment. Before fine-tuning, Llama3's results were often completely unrelated.

BehnamGhader et al., 2024) contrastive learning loss, where the original captions and re-annotated captions generated by ShareCaptioner (Chen et al., 2023) were treated as positive pairs, pulling them closer. In contrast, all other captions formed a negative sample set that the model learned to push away. Remarkably, after this CC fine-tuning, the caption retrieval accuracy, as shown in Table 1, rose from 18.4% to 73%, which is a 7% improvement over the original CLIP-ViT-L text encoder. This successful fine-tuning process enables us to more effectively harness the open-world capabilities of LLMs for CLIP training.

In a nutshell, we present *LLM2CLIP*, a novel approach for enhancing visual representation learning through the integration of large language models (LLMs) as shown in Figure 1. This method takes a straightforward yet audacious step by replacing the original CLIP text encoder and augmenting the CLIP visual encoder with the vast knowledge embedded in LLMs. We have identified key obstacles associated with this innovative idea and proposed a cost-effective fine-tuning strategy to overcome them. Our experiments demonstrate that leveraging LLMs as teachers for CLIP training yields substantial improvements, with *LLM2CLIP* significantly outperforming state-of-the-art pre-trained CLIP models. Our method increased the performance of the previously SOTA EVA02 model by 16.5% on both long-text and short-text retrieval tasks, transforming a CLIP model trained solely on English data into a state-of-the-art cross-lingual model. Furthermore, when incorporated into multimodal model training, such as with Llava 1.5, it consistently achieved comprehensive improvements over EVA02 across nearly all benchmarks. Additionally, the efficient training method proposed by *LLM2CLIP* ensures that the training cost is nearly identical to fine-tuning the original CLIP. We have also demonstrated that using more powerful language models and larger training datasets can further boost *LLM2CLIP*'s performance, showcasing the immense potential of our approach. These promising outcomes affirm that we have successfully transformed CLIP into a more general-purpose foundational model. The enhanced *LLM2CLIP* model possesses richer knowledge and exhibits a remarkable capacity for distinguishing fine-grained and complex long-text semantics. This advancement not only broadens the range of supported downstream tasks but also propels progress across the entire vision domain.

## 2 RELATED WORKS

**Contrastive Language-Image Pre-training (CLIP).** CLIP (Radford et al., 2021) is one of the most important multimodal models today. By training on web-scale data, image captions serve as rich, fine-grained supervision signals for learning representations that align closely with image, freeing CLIP from the limitations of manually defined labels. The richness of the captions endows CLIP with many remarkable capabilities; besides zero-shot image-text retrieval, CLIP features can even achieve region-level correspondence, supporting zero-shot detection and segmentation tasks. Furthermore, CLIP is the most widely used encoder in current image and video understanding tasks. In Tong et al. (2024)'s analysis, it stands out among various visual encoders in MLLMs training as the most significant model, demonstrating its strong capabilities. For generative tasks, CLIP is equally foundational. Its text and image encoders are used to provide text control (Rombach et al., 2021; Ramesh et al., 2022) and image control signals (Blattmann et al., 2023; Zhang et al., 2023) for conditioning generative models. Additionally, the CLIP score (Hessel et al., 2021) has become a crucial metric for assessing the relevance of generated images to text. While the original CLIP only utilized image and text modalities, there have been massive extensions of CLIP to incorporate various other modalities (Dai et al., 2024; He et al., 2023; Moon et al., 2022; Zhang et al., 2022). CLIP definitely has a significant impact on the field of vision and multimodal, and its influence is still growing.

**CLIP meets Stronger Language Models.** Several works have explored the integration of LLMs into CLIP. JinaCLIP (Koukounas et al., 2024) employed Jina-embeddings-v2 (Günther et al., 2023) as the text encoder, which is a BERT variant with 137M parameters, supporting longer texts. Though achieving similar visual performance to EVA-CLIP (Sun et al., 2023), its text encoder is far behind our used Llama3-8B, limiting the potential benefits from LLMs. T5-V (Jiang et al., 2024) took a different approach by aggregating LLM features at the MLLM layer of LLAVA-next-8B (Li et al., 2024a) while freezing the ViT gradient to focus on adjusting the MLLM output. However, freezing visual encoder does not address the problem that ViT inherently lack complex visual feature extraction capabilities, leading to much worse performance than LLM2CLIP. MATE (Jang et al., 2024) designed a learnable adaptor to bridge the gap between CLIP's text encoder and LLMs. They trained the CLIP visual encoder using LoRA on a small dataset focused on long-text image retrieval tasks. However, they did not recognize the critical issue we propose in this paper: the poor separability of LLM feature space, which is insufficient for direct support of CLIP training. Our work aims to thoroughly explore the impact of LLMs on CLIP as a foundational model, going beyond the influence on long-text retrieval.

**CLIP meets Longer Captions.** It has been widely recognized that the quality of CLIP's text embedding is coarse and limited to only 77 tokens. Many works have attempted to extend the length of CLIP captions and retrain CLIP accordingly. For instance, DCI (Urbanek et al., 2024) hired annotators to expand captions based on SAM (Kirillov et al., 2023)'s target hints. LaCLIP (Fan et al., 2024) adopted ChatGPT, Bard, and human rewriting. DreamLIP (Zheng et al., 2024) leveraged ShareCaptioner (Chen et al., 2023) and InstructBLIP to augment 30M captions. Recap-DataComp-1B (Li et al., 2024b) used Llama3-trained LLAVA1.5 to extend 1B captions. In order to handle captions longer than 77 tokens, these methods compromised by summarizing long captions into shorter ones (Urbanek et al., 2024), splitting captions into multiple segments (Fan et al., 2024; Zheng et al., 2024), or finetuning model's positional encoding to support longer token inputs (Zhang et al., 2024). LLM2CLIP, in comparison, leverages Llama3 as the text encoder, which enables comprehensive understanding of long and dense captions. It not only resolves the limition of token length but also allows for better understanding using LLMs' open-world knowledge. As a result, our model achieves a breakthrough in performance.

## 3 METHODS

The contributions of our methodology are threefold: First, we designed experiments to analyze the key reason preventing LLMs from directly participating in multimodal representation learning — the weak discriminability of their output features. Second, we introduced the caption contrastive fine-tuning method, which significantly improves feature discriminability. Third, we developed the *LLM2CLIP* training framework, which has been proven to be an efficient and effective method for leveraging LLMs to deliver substantial performance improvements to pretrained CLIP models.

### 3.1 NATIVE LLMS ARE INEFFECTIVE TEXT ENCODERS FOR CLIP

Large Language Models (LLMs), such as Llama-3, are trained on massive world corpora through auto-regression, equipping them with open-world knowledge and enabling them to perform various tasks. We initially hoped to leverage the capabilities of LLMs to directly retrain a CLIP model.

Although LLMs exhibit strong text comprehension, they are difficult to use directly as text embedding models. This is because their knowledge is encapsulated within the model, and their output features are heavily skewed towards individual word predictions. As generative models, they are not trained to ensure good linear separability of output features, making them less effective when used to interpret captions for CLIP. As highlighted by Chen et al. (2022), cross-modal contrastive learning in CLIP requires that each modality possesses strong internal discriminability.

To evaluate the effectiveness of various language models in terms of text discriminability, and to test whether the native output features of LLMs, as hypothesized, struggle to distinguish image captions, we introduce a new metric: the MS COCO Caption Retrieve Accracy (CRA). MS COCO is a widely used multimodal dataset, containing over 330K images, each with five captions. These captions are written by different human annotators and provide diverse descriptions for each image. In our evaluation on the MS COCO validation set, we use only the first two captions of each image and treat the captions of the same image as positive pairs, while all other captions serve as negative samples. We then perform caption-to-caption retrieval and assess Top-1 accuracy using different language models, defining the result as their CRA score. Higher CRA scores indicate better discriminability of the language models on image captions.

As shown in Table 1, using a pure LLM results in a CRA score of only 18.4%, indicating that the majority of captions cannot be well-separated in the output space. In fact, captions with similar distances may be entirely unrelated, as illustrated in Figure 2. However, the text encoder from the original state-of-the-art CLIP model achieves a CRA score of 66%, proving the inadequacy of native LLM output features in caption discriminability. Consequently, it is challenging to apply LLMs directly in CLIP model training. Subsequent experiments in Ta-

ble 6 also confirm that replacing CLIP's text encoder and the corresponding ViT with Llama-3 8B for contrastive learning significantly underperforms the original CLIP.

## 3.2 A Crucial Lesson: LLM Learning for Image Caption Discrimination

In this section, we aim to fine-tune the LLM's token outputs to better capture features that can distinguish between image captions.

The process of improving the discriminability of LLM output features on caption text is quite straightforward: we need the distances between captions of the same image to be closer, and those of different images to be further apart. Therefore, we apply a caption contrastive (CC) fine-tuning to the LLM's output features, treating different captions of the same image as positive samples and the rest of the captions as negative samples.

To obtain enough varied descriptions for the same image, we use the ShareCaptioner (Zheng et al., 2024; Chen et al., 2023) modified CC-3M (Sharma et al., 2018) dataset, which provides both original captions and augmented dense captions for each image. These can be treated as positive pairs. We followed the training methodology of LLM2Vec (BehnamGhader et al., 2024), first expanding the LLM's attention mechanism to bidirectional attention, and employing Masked Next Token Prediction (MNTP) for initialization to achieve better results. Specifically:

First, we transform the LLM's causal attention mechanism into bidirectional attention, as we no longer need it to retain generative capabilities but rather function as an encoder. Since autoregressive training is no longer required, switching to bidirectional attention improves its ability to capture contextual information. Second, we employ MNTP to train the newly added bidirectional attention mechanism, providing a strong initialization. For a given sequence of $N$ tokens, we mask a subset and predict their values, similar to BERT (Devlin et al., 2018). However, unlike BERT, we adapt this process to fit the nature of LLMs by predicting tokens just before the masked token. We train on both image captions and pure text with equal weighting. In addition to CC-3M, we also use the Wikitext-103 (Merity et al., 2016) dataset to preserve the LLM's text capabilities and prevent divergence from its original strengths. Finally, we perform the actual caption contrastive fine-tuning, using a supervised SimCSE loss to bring captions of the same image closer together and push apart captions of different images. We use two prompt templates: "Given a caption, retrieve a detailed relevant caption" and "Given a detailed caption, retrieve a short relevant caption," which are prepended to the query (either original or dense caption) to retrieve the corresponding dense or original caption. Similarly, we use a 1.5M-paired general text dataset curated from Springer et al. (2024) to maintain strong performance in pure language tasks. All training is efficiently conducted using LoRA and is completed in just one epoch, ensuring low computational cost.

A remarkable result followed: after fine-tuning the LLM as a caption encoder for only one epoch on CC-3M, the CRA score for Llama-3 8B jumped from 18.4% to 73.0%, surpassing the previous state-of-the-art CLIP and EVA models' text encoders trained on the massive Laion-2B (Schuhmann et al., 2022) and COYO-700M (Byeon et al., 2022) image-text datasets. As shown in Table 6, subsequent experiments demonstrated that after CC fine-tuning, the LLM finally unleashed its powerful capabilities, significantly boosting the performance of the previously state-of-the-art CLIP model, in stark contrast to the results without CC fine-tuning. *This breakthrough uncovers the potential of LLMs in CLIP training and removes a major obstacle in leveraging LLMs to advance vision foundation models.*

## 3.3 *LLM2CLIP*: Makes SOTA CLIP more SOTA ever

With the modifications to the LLM discussed earlier, we have now obtained a super text encoder that is well-suited for CLIP training. The next step is to use this LLM in conjunction with the pretrained state-of-the-art CLIP visual encoder to reconstruct a more powerful cross-modal feature space.

As shown in Figure 1, in the *LLM2CLIP* training phase, we freeze the gradients of the LLM to preserve its inherent capabilities, for two primary reasons. First, this significantly reduces the computational cost and memory footprint of fine-tuning. CLIP training requires a very large batch size to maintain the effectiveness of negative samples. Allocating memory to the LLM could compromise CLIP's performance. Second, by freezing the LLM, we ensure that the open-world knowledge it has acquired from large-scale corpora remains intact during the multimodal alignment process.

To compensate for the frozen LLM, and inspired by methods like FuseMix (Vouitsis et al., 2023) and APE (Rosenfeld et al., 2022), we introduce several new linear layers after the LLM as adapters. These layers serve as learnable parameters to improve the alignment between the LLM and CLIP visual encoder. Following the original design of CLIP, we also employ a projector layer to align the dimensions of the two encoders, facilitating the use of CLIP loss for training.

With this powerful LLM-based super text encoder, we achieve a substantial qualitative leap in CLIP's language comprehension capabilities. The LLM's open-world knowledge allows the CLIP visual encoder to learn more structured and globally informed visual representations that are aligned with human knowledge. Moreover, this

approach enables us to fully utilize high-quality, long, and dense caption datasets without requiring any special architectural adjustments, which previous works like DCI, DreamLip, and Recaption struggled to effectively leverage. As shown in Table 2, *LLM2CLIP* makes any existing SOTA CLIP model even more SOTA, significantly surpassing the performance of previous one.

### 3.4 OVERVIEW AND EFFICIENCY DISCUSSION

We propose *LLM2CLIP* as a method that efficiently incorporates large language models (LLMs) into CLIP training, leveraging the capabilities of LLMs to make cross-modal representation learning significantly more powerful. In our experiments, We evaluated large language models, including Llama with 1B and 8B parameters, as well as Mistral-Nemo with 12B parameters.It might seem that incorporating such large LLMs would greatly increase the computational burden of training CLIP, especially since CLIP itself is computationally expensive, requiring a large batch size. However, our proposed *LLM2CLIP* is remarkably lightweight. The training overhead is **nearly identical** to fine-tuning the original CLIP model, with minimal additional cost, yet the LLM provides much stronger supervision.

Here, we highlight some of the design details that significantly improve training efficiency: 1). During the caption contrastive fine-tuning stage, we employ LoRA training for the LLM. Even for a 12B LLM, training with a batch size of 512 only requires around 70GB of GPU memory, making it feasible to run on a single 80GB 8 A100 GPU node. 2). In the *LLM2CLIP* stage, we freeze the LLM's gradients and only train the learnable adapter, CLIP's original Vision encoder, and two projectors. The additional trainable parameters are roughly equivalent to those in the original CLIP, minimizing the overhead. To further reduce the inference cost of using LLMs, we pre-extract all the text features from the training data and store them in memory. This ensures that, even though LLMs provide powerful textual supervision, the memory and computational costs during training remain **nearly identical** to those of standard CLIP training.

For instance, when we trained *LLM2CLIP* using a Mistral-Nemo[1] 12B model integrated with the commonly used EVA ViT-L/14-224, with a batch size of 4096 on 8 H100 GPUs, the memory usage per GPU was only 30GB, and the entire training process took just 9 hours. Despite this efficient training cost, *LLM2CLIP* brought transformative improvements in downstream tasks such as long and short text retrieval, cross-language retrieval, and LLAVA training.

## 4 EXPERIMENTS

### 4.1 EXPERIMENT SETTING

#### 4.1.1 CAPTION CONTRASTIVE FINE-TUNING

**Training Dataset.** We utilized the ShareCaptioner-modified CC-3M dataset (Zheng et al., 2024; Chen et al., 2023), which provides both original captions and augmented dense captions for each image, for contrastive learning. For the Masked Next Token Prediction and caption contrastive fine-tuning stages, we employed the Wikitext-103 dataset (Merity et al., 2016) and the E5 dataset from Springer et al. (2024) to ensure that the general text domain was not excessively biased.

**Training Setting.** We trained the language model using LoRA, applying lightweight training with 1 epoch on all datasets. We adopted the average of all LLM output tokens as the text global embedding for a caption. All training parameters follow the design of BehnamGhader et al. (2024).

#### 4.1.2 *LLM2CLIP* FINE-TUNING

**Training Dataset.** To compare different configurations, we designed three experimental settings based on dataset size: *LLM2CLIP*-3M: This configuration only uses the CC-3M dataset for training, representing our lightweight setting. *LLM2CLIP*-15M: This configuration uses both CC-3M and CC-12M datasets, which is our default setting. All *LLM2CLIP* models without a specific dataset size label are trained with this 15M data. *LLM2CLIP*-60M: This configuration scales up the dataset, combining CC-3M, CC-12M, YFCC-15M, and a randomly selected 30M subset from Recaption-1B (Li et al., 2024b). All data used are dense captions rewritten by multimodal large language models (MLLMs). The CC-3M, CC-12M, and YFCC datasets are sourced from Zheng et al. (2024) using the ShareCaptioner for rewriting, while Recaption data was rewritten using Llama3-LLAVA1.5. We primarily used a mix of original captions and dense captions, with the default mixing ratio being 1:1. *LLM2CLIP*-S represents a setting where only original data is used for training, matching the original CLIP pretraining distribution to analyze the model's benefits separately.

---

[1] https://mistral.ai/news/mistral-nemo/

**Training Setting.** All experiments with different datasets were conducted for 4 epochs. We froze the LLM gradients by default. During training, we pre-extracted the text features from the captions using the LLM and stored them in memory to avoid repeated LLM inference, reducing additional computational overhead. We trained the original CLIP vision encoder, the LLM's learnable adapter, and the projectors for both encoders. Training on the 3M, 15M, and 60M datasets for CLIP ViT-L/14-224 required only 30G per GPU memory and 2, 9, and 45 hours on 8 H100 GPUs, respectively.

### 4.1.3 EVALUATION

**Evaluation Dataset.** For short-text datasets, we used the commonly available COCO 2014 5k test set and the Flickr 1k test set as our test datasets. For long-text datasets, we used the 1K subset of the ShareGPT4V (Chen et al., 2023) dataset and the Urban1k (Zhang et al., 2024) dataset, both provided by LongCLIP, along with the DOCCI (Onoe et al., 2024) dataset. The ShareGPT4V-1M dataset was generated using annotations from GPT-4V and ShareCaptioner, with images from Laion, CC, SBU (Ordonez et al., 2011) and MS COCO. We used a randomly selected 1K subset of this dataset. Urban1k consists of captions generated by GPT-4V for 1,000 busy urban view images from Visual Genome. Each caption is a long and complete sentence that describes the image, including types, colors, and relative locations of various attributes. The model can only successfully match the images with the correct captions if it accurately understands and models the detailed attributes in both modalities. The DOCCI dataset contains approximately 1.5K high-resolution images with detailed human-annotated descriptive captions. DOCCI is divided into a training set of 9.6K pairs and a test set of 5.1K pairs. We used the test set for image-lengthy caption retrieval experiments. For the Chinese retrieval tasks, we used the FlickrCN (Lan et al., 2017) and CNCOCO (Li et al., 2018) datasets, which are translated versions of Flickr30K and MS COCO-1K, respectively. These datasets were tested in Chinese. For Llava (Liu et al., 2023a) training, we used the standard training and test sets from LLAVA 1.5.

**Evaluation Setting.** The experiments in this paper primarily focus on how *LLM2CLIP* enhances the performance of widely used vanilla CLIP across various dimensions. We mainly compare our approach with the EVA (Fang et al., 2023) and OpenAI CLIP models as baselines, as they are the most widely used SOTA vision encoders in the open-source community. The CLIP models we use for comparison are ViT-B/16[2], ViT-L/14[3], and ViT-L/14-336[4]. For EVA02 models, we use EVA02 ViT-B/16[5], EVA02 ViT-L/14[6], and EVA02 ViT-L/14-336[7], making up a total of six models. For language models, we experimented with four different models: Jina-Embeddings-V2[8], and three popular LLMs from the Llama3 (Dubey et al., 2024) family, namely Llama 3.2 1B, Llama 3 8B, and Mistral-Nemo 12B. To avoid any misunderstanding, the experiments in this paper, unless otherwise noted, use EVA02 ViT-L/14 as the default baseline for comparison. The default language model used is LLaMA 3 8B, trained on a dataset of 15M setting mentioned above.

### 4.2 MAIN RESULTS

**Directly Replacing CLIP's Text Encoder with a Vanilla LLM is Harmful.** As shown in the experiments in Table 4, directly replacing the text encoder of EVA02 ViT-L/14 with Llama3-8B significantly degrades retrieval performance. For example, performance on the DOCCI benchmark nearly halves, dropping from 75.0/73.4 to 51.7/50.6. Additionally, the CRA score of the vanilla Llama3-8B is particularly low, indicating that its output features exhibit very poor discriminability for captions. These results clearly show that such an approach imposes a significant burden on CLIP's learning process.

**Improving the Discriminability of LLM Output Features is Key to Integrating LLMs with CLIP.** To enhance the discriminability of LLM output features, we applied Caption Contrastive fine-tuning. Llama3-8B-TC and Llama3-8B-CC represent the results of using pure text corpora and a mix of text and augmented CC3M corpora from ShareCaptioner, respectively, both trained with supervised SimCSE loss. As shown in the Table 6, the contrastive learning on mixed caption corpora yields higher CRA scores compared to training on generic text, with noticeable improvements across almost all benchmarks. This difference stems from the fact that LLMs trained on caption-distributed data have stronger discriminability for caption features. These findings highlight the importance of text model discriminability for CLIP training and underscore the rationale behind the caption contrastive fine-tuning in *LLM2CLIP*.

---

[2] https://huggingface.co/openai/clip-vit-base-patch16
[3] https://huggingface.co/openai/clip-vit-large-patch14
[4] https://huggingface.co/openai/clip-vit-large-patch14-336
[5] https://huggingface.co/QuanSun/EVA-CLIP/blob/main/EVA02_CLIP_B_psz16_s8B.pt
[6] https://huggingface.co/QuanSun/EVA-CLIP/blob/main/EVA02_CLIP_L_psz14_s4B.pt
[7] https://huggingface.co/QuanSun/EVA-CLIP/blob/main/EVA02_CLIP_L_336_psz14_s6B.pt
[8] https://huggingface.co/jinaai/jina-embeddings-v2-base-en

Table 2: Systematic Comparison Experiment Demonstrating the Performance Improvements of LLM2CLIP.

| Methods | Flickr30k | | COCO | | ShareGPT4V | | Urban-1k | | DOCCI | |
|---|---|---|---|---|---|---|---|---|---|---|
| | I2T | T2I | I2T | T2I | I2T | T2I | I2T | T2I | I2T | T2I |
| **ViT-B/16** | | | | | | | | | | |
| ALIGN | 80.6 | 62.2 | 52.0 | 43.2 | 75.9 | 80.6 | 62.2 | 59.1 | 59.7 | 62.1 |
| BLIP | 80.6 | 74.1 | 61.7 | 48.5 | 65.8 | 74.3 | 45.5 | 48.5 | 50.5 | 53.5 |
| Jina-CLIP | 80.6 | 67.4 | 55.6 | 41.1 | - | - | 87.7 | 88.0 | 78.7 | 80.0 |
| Long-CLIP | 85.8 | 70.6 | 56.9 | 40.9 | 94.8 | 93.5 | 79.1 | 79.1 | 63.1 | 71.4 |
| CLIP | 82.3 | 62.2 | 52.4 | 33.1 | 84.5 | 79.8 | 67.5 | 53.1 | 60.7 | 57.1 |
| *+LLM2CLIP* | **9.2** | **78.1** | 62.2 | 48.7 | 98.1 | 97.4 | **86.1** | **90.0** | 84.1 | 85.0 |
| EVA02 | 86.2 | 71.5 | 58.7 | 42.1 | 90.5 | **85.5** | 67.0 | 60.8 | 67.7 | 68.0 |
| *+LLM2CLIP* | 88.5 | 78.0 | **63.6** | **49.8** | **98.0** | 98.1 | 84.7 | 89.7 | **85.5** | **86.8** |
| **ViT-L/14** | | | | | | | | | | |
| Long-CLIP | 90.0 | 76.2 | 62.8 | 46.3 | 97.2 | 97.3 | 82.5 | 86.1 | 66.5 | 78.6 |
| CLIP | 85.2 | 65.0 | 56.3 | 36.5 | 84.2 | 83.6 | 68.3 | 55.6 | 63.1 | 65.8 |
| *+LLM2CLIP* | 92.6 | 81.7 | 64.9 | 52.5 | 98.4 | 98.4 | 87.6 | 92.0 | 87.6 | 88.7 |
| EVA02 | 89.7 | 77.3 | 63.7 | 47.5 | 91.9 | 89.3 | 73.3 | 68.5 | 73.5 | 75.0 |
| *+LLM2CLIP*-3M | 89.6 | 77.3 | 59.7 | 48.0 | 98.3 | 98.6 | 87.1 | 91.1 | 84.9 | 87.8 |
| *+LLM2CLIP* | 92.0 | 82.8 | 68.5 | 54.8 | 98.6 | 99.0 | 88.1 | 94.0 | 88.2 | 90.4 |
| *+LLM2CLIP*-30M | 92.0 | **83.5** | 69.0 | 55.3 | 98.9 | 98.8 | 93.1 | 95.0 | 89.3 | 91.2 |
| *+LLM2CLIP*-60M | **94.4** | 83.2 | **70.4** | **55.7** | **99.2** | **99.4** | **94.1** | **95.2** | **90.2** | **92.0** |
| **ViT-L/14-336** | | | | | | | | | | |
| CLIP | 87.7 | 67.0 | 58.0 | 37.1 | 86.2 | 84.0 | 72.8 | 57.0 | 67.4 | 65.7 |
| *+LLM2CLIP* | 91.2 | 82.1 | 65.5 | 53.6 | 98.1 | 98.4 | 90.3 | 93.2 | 87.7 | 89.0 |
| *+LLM2CLIP*-60M | 93.9 | 82.3 | 68.5 | 54.8 | **98.9** | 99.1 | **94.6** | **95.9** | **89.6** | 90.6 |
| EVA02 | 89.6 | 78.0 | 64.2 | 47.9 | 91.5 | 89.4 | 76.6 | 70.0 | 74.7 | 76.4 |
| *+LLM2CLIP* | **93.9** | **83.8** | **68.7** | **55.7** | 98.8 | **99.2** | 89.5 | 94.2 | 89.2 | **91.3** |

Table 3: Retrieval Performance across Flickr30K-CN and COCO-CN.

| Methods | Flickr-CN | | | | | | COCO-CN | | | | | |
|---|---|---|---|---|---|---|---|---|---|---|---|---|
| | I2T@1 | I2T@5 | I2T@10 | T2I@1 | T2I@5 | T2I@10 | I2T@1 | I2T@5 | I2T@10 | T2I@1 | T2I@5 | T2I@10 |
| **ViT-L/14-336** | | | | | | | | | | | | |
| Wukong | 76.1 | 94.8 | 97.5 | 51.7 | 78.9 | 86.3 | 53.4 | 80.2 | 90.1 | 55.2 | 81.0 | 90.6 |
| CN-CLIP | 80.2 | 96.6 | 98.2 | 68.0 | 90.7 | 95.4 | 63.4 | 84.2 | 92.9 | 64.0 | 89.2 | 94.4 |
| JinaCLIP | 3.30 | 9.90 | 15.1 | 0.7 | 3.5 | 6.0 | 2.9 | 8.9 | 13.7 | 1.0 | 4.9 | 8.2 |
| EVA02 | 4.40 | 11.8 | 16.7 | 0.94 | 2.9 | 4.8 | 2.7 | 9.8 | 15.2 | 1.0 | 3.7 | 7.3 |
| *+LLM2CLIP* | **86.9** | **98.1** | **99.3** | **75.1** | **92.9** | **96.0** | **69.1** | **92.5** | **97.2** | **70.0** | **92.6** | **96.7** |

***LLM2CLIP* Makes Pretrained SOTA CLIP Even More SOTA.** We applied the *LLM2CLIP* fine-tuning method to both EVA02 and CLIP models and observed significant performance improvements in Table 2. Even with lightweight fine-tuning, the results substantially surpassed those of the original models pretrained on datasets like Laion2B. Compared to other methods such as LongCLIP and JinaCLIP, which also attempt to fine-tune pretrained CLIP models, our performance gains were transformative. This demonstrates the effectiveness of *LLM2CLIP* as a method for introducing LLMs to enhance CLIP's performance.

Table 4: Ablation Study of LLM2CLIP. Here LLM2CLIP-S refers to the results trained on the original short caption dataset.

| Methods | Flickr30k | | COCO | | ShareGPT4v | | Urban-1k | | DOCCI | |
|---|---|---|---|---|---|---|---|---|---|---|
| | I2T | T2I | I2T | T2I | I2T | T2I | I2T | T2I | I2T | T2I |
| **EVA02 Vit-L/14** | 89.7 | 77.3 | 63.7 | 47.5 | 91.9 | 89.3 | 73.3 | 68.5 | 73.5 | 75.0 |
| + Jina-Bert | 88.1 | 77.7 | 60.5 | 51.1 | 83.3 | 81.0 | 66.9 | 68.5 | 68.9 | 71.2 |
| ++ Dense Caption | 87.9 | 77.9 | 60.9 | 50.3 | 95.3 | 95.1 | 79.4 | 83.8 | 73.8 | 77.9 |
| + Llama3-8B-S | 87.9 | 75.6 | 56.7 | 41.8 | 55.1 | 46.1 | 37.2 | 35.1 | 39.3 | 32.3 |
| ++ CC Finetuning | **92.4** | **82.9** | 67.6 | 54.5 | 97.7 | 94.9 | 75.8 | 83.4 | 83.7 | 85.6 |
| +++ Dense Caption | 92.0 | 82.8 | **68.5** | **54.8** | **98.6** | **99.0** | **88.1** | **94.0** | **88.2** | **90.4** |

**LLM's Built-in Knowledge Directly Benefits CLIP.** In Table 4's ablation study, we examine the impact of different steps in *LLM2CLIP* on the EVA02 model. When using only the original captions, there was minimal improvement, as these captions are from the same distribution as those used in EVA02 CLIP's pretraining. However, we observed that the LLM, through its open-world knowledge, is able to provide a deeper understanding of the original captions, empowering the CLIP model with enhanced capabilities and further boosting performance. Naturally, when dense captions are introduced, the performance improvement becomes even more pronounced.

**LLM Improves CLIP's Ability to Handle Long and Dense Captions.** Thanks to the LLM's inherent long context windows and strong text understanding capabilities, CLIP can process long caption datasets more ef-

fectively without requiring any architectural modifications. This stands in contrast to methods like LongCaption, which require fine-tuning of positional encodings, or Dreamlip, which splits captions into sub-captions. Jina CLIP, for example, enables long-text input by replacing its text encoder with Jina BERT. To verify the LLM's natural advantage with long texts, we compared our method to Jina CLIP to assess whether the LLM provides a better understanding of long captions. As shown in Table 4, when dense captions are added during training, *LLM2CLIP* demonstrates significantly greater performance improvements than Jina CLIP, proving that LLMs are better suited to processing long and dense captions.

**Larger LLMs Lead to Better Performance.** LLM2CLIP explores the method of replacing CLIP's text encoder with an LLM and continuing training. In Table X, we thoroughly investigate the performance differences that various versions of LLMs bring to CLIP. In the experiments shown in Table 6, we swapped the text encoder in CLIP with different language models to explore their impact. We maintained the original LLM2CLIP design, with the only change being the initialization of the text encoder. As expected, within the same Llama family, the 8B model significantly outperforms the 1B model. Mistral Namo also shows a slight improvement over the 8B model. It's important to note that during CLIP training, the LLM's gradients remain frozen, further proving that larger LLMs, with their richer knowledge, can deliver substantial performance improvements to CLIP. Even so, the 1B LLaMA3.2 model still achieved an impressive performance boost for EVA02 CLIP, with an average improvement of 11.1% — an impressive feat considering that EVA02 is already one of the SOTA open-source CLIP models.

Table 5: **Comparison Experiment of Different Ratios of Dense Captions in the LLM2CLIP Training Process.**

| Ratio | Flickr30k | | COCO | | ShareGPT4v | | Urban-1k | | DOCCI | |
|---|---|---|---|---|---|---|---|---|---|---|
| | I2T | T2I | I2T | T2I | I2T | T2I | I2T | T2I | I2T | T2I |
| **100%** | 85.5 | 72.7 | 60.1 | 46.9 | **98.7** | 99.0 | 88.7 | 93.9 | 88.0 | **90.5** |
| **75%** | 92.4 | 82.6 | 68.5 | 54.2 | **98.7** | **99.3** | **89.0** | **94.3** | 88.1 | 90.2 |
| **50%** | 92.0 | 82.8 | **68.5** | **54.8** | 98.6 | 99.0 | 88.1 | 94.0 | **88.2** | 90.4 |
| **25%** | **93.0** | 82.8 | 68.1 | 54.8 | 98.4 | 98.7 | 87.7 | 92.9 | 87.9 | 90.0 |
| **0%** | 92.4 | **82.9** | 67.6 | 54.5 | 97.7 | 94.9 | 75.8 | 83.4 | 83.7 | 85.6 |

Table 6: Comparison of various text encoders.

| Methods | Flickr30k | | COCO | | ShareGPT4v | | Urban-1k | | DOCCI | | Average | CRA |
|---|---|---|---|---|---|---|---|---|---|---|---|---|
| | I2T | T2I | I2T | T2I | I2T | T2I | I2T | T2I | I2T | T2I | | |
| **EVA02 Vit-L/14** | 89.8 | 73.3 | 63.8 | 63.8 | 89.3 | 91.9 | 68.5 | 73.3 | 75.0 | 73.4 | 76.2 | 69.8 |
| **+Jina Bert** | 87.9 | 77.9 | 60.9 | 50.3 | 95.3 | 95.1 | 79.4 | 83.8 | 73.8 | 77.9 | 78.2 | **74.2** |
| **+Llama3-8B** | 87.1 | 75.3 | 56.4 | 41.6 | 89.3 | 91.4 | 58.6 | 60.9 | 51.7 | 50.6 | 66.3 | 18.4 |
| **+Llama3-8B-TC** | 92.7 | 82.1 | 68.1 | 54.6 | 97.7 | 98.2 | 88.9 | 93.8 | 85.0 | **97.8** | 84.8 | 71.3 |
| **+Llama3-8B-CC** | 92.0 | 82.8 | 68.5 | **54.8** | 98.6 | 99.0 | 88.1 | 94.0 | **88.2** | 90.4 | 85.6 | 73.0 |
| **+Llama3.2-1B-CC** | 91.6 | 81.3 | 65.8 | 52.5 | 98.3 | 98.2 | 84.5 | 91.9 | 83.4 | 86.4 | 83.4 | 72.8 |
| **+Mistral-Nemo-12B-CC** | **93.5** | **83.7** | 68.5 | 54.7 | **98.6** | 98.9 | **90.4** | **94.3** | 88.0 | 89.7 | **86.0** | 73.3 |

Table 7: **Performance of Llava 1.5.** The best results are highlighted in **bold**. We explored whether LLM2CLIP could enhance complex image understanding tasks by modifying Llava's visual encoder. The results showed significant improvements across most tasks, demonstrating the ability of LLM2CLIP to effectively enhance visual understanding in more complex scenarios.

| MODEL | VQA Datasets | | | | | Pope Metrics | | | MM Benchmarks | | | | | Seed Benchmarks | | |
|---|---|---|---|---|---|---|---|---|---|---|---|---|---|---|---|---|
| | VQAv2 | GQA | VizWiz | SQA-IMG | TextVQA | Random | Adv. | Popular | MME | MMBench | MMBench-CN | LlavaBench | MMVet | All | IMG | Video |
| Llava (Paper) | 78.5 | 62.0 | 50.0 | 66.8 | 58.2 | 87.3 | 86.1 | 84.2 | 1510.7 | 64.3 | 58.3 | 65.4 | 31.1 | 58.6 | 66.1 | 37.3 |
| Llava (Rep.) | 79.04 | **62.86** | 50.57 | 67.97 | 57.48 | **87.7** | **84.85** | 86.3 | 1476.69 | 66.66 | **60.39** | 58.0 | 34.3 | 59.86 | 66.95 | **39.71** |
| *+LLM2CLIP* | **79.68** | 62.79 | **52.24** | **69.61** | **58.19** | 87.62 | 84.63 | **86.6** | **1483.15** | **68.47** | 59.19 | **62.7** | **34.8** | **60.21** | **67.6** | 39.21 |

**Like a miracle: LLM enables English CLIP to learn Chinese without exposure to Chinese data.** To validate the knowledge transfer capabilities of the LLM, we designed an out-of-distribution (OOD) experiment by conducting an image-text retrieval task in a completely unfamiliar language. This is an especially challenging experiment because all of our training was performed on purely English text, yet now we are testing the model on Chinese data. As shown in Table 3, models like EVA02 CLIP and Jina CLIP, which performed well on English datasets, achieved near-zero accuracy on this task. However, the magical power of *LLM2CLIP* became evident: the LLM's capabilities allowed the model to achieve impressive performance on Chinese retrieval tasks, even surpassing models that were originally trained on hundreds millions of Chinese data, such as Wukong (Gu et al., 2022) and CN-CLIP (Yang et al., 2022). This result once again demonstrates that *LLM2CLIP* can effectively integrate the inherent abilities of the LLM into CLIP, enabling it to handle tasks far beyond its original scope.

## 4.3 ABLATION

**Impact of Different Data Ratios.** With the integration of an LLM, *LLM2CLIP* enhances our ability to understand dense captions. In Table 5, we present an ablation study that examines the impact of different ratios of dense captions versus original captions on CLIP's training performance. The "Ratio" column represents the proportion of dense captions used during training. Overall, dense captions contribute to a noticeable improvement in performance, but more is not always better. For example, when using 0% dense captions, the performance on datasets like COCO and Flickr30K remains decent, but long-text benchmarks such as ShareGPT4V, Urban-1K, and DOCCI show poor results. Conversely, with 100% dense captions, performance on short-text retrieval benchmarks is the worst. It's important to note that the dense captions we used were generated by the ShareCaptioner model, and there may be some distribution differences and noise compared to real data captions, which could somehow explain why using 100% dense data is suboptimal. Our findings indicate that the best performance is achieved when dense captions make up 50% to 75% of the training data, striking a balance between both short-text and long-text retrieval tasks.

**Impact of Different Data Size.** As shown in Table 2, larger training datasets consistently yield positive results for LLM2CLIP, with clear improvements in both long-text and short-text retrieval tasks. Our 60M dataset version has already pushed the limits of what CLIP could previously achieve. Even the 3M version, though lightweight, still delivers significant performance gains, demonstrating the efficiency of the LLM2CLIP approach. It's worth noting that training on the 3M dataset takes only about 3 hours on an 8 H100 GPUs machine, yet results in a transformative leap in performance for a well-pretrained CLIP ViT model.

## 5 LIMITATIONS AND FUTURE WORK

This paper focuses solely on the impact of LLMs on CLIP. In reality, there are many other areas where text can play a transformative role, such as diffusion models (Rombach et al., 2021; Blattmann et al., 2023; Tech, 2023), Segment Anything (Kirillov et al., 2023) and GroundingDINO (Liu et al., 2023b), both of which could potentially see significant performance improvements by applying our method.

Additionally, while LLMs possess open-world knowledge, aligning them with CLIP may require specific data to fully unlock their potential. LLM2CLIP is a method that requires only a small amount of data for fine-tuning in conjunction with a pretrained CLIP visual encoder, and we have already demonstrated significant performance gains. However, we did not intentionally select fine-tuning data based on specific characteristics. By focusing on aspects such as data distribution, length, or categories, we could further tailor the LLM to address CLIP's limitations, allowing the LLM to act as a more comprehensive teacher for various tasks.

Moreover, in this work, we froze the gradients of the LLM during fine-tuning to maintain a large batch size for CLIP training. However, it would be worth exploring approaches that allow for updating the LLM's gradients, as there may be better engineering practices that can balance the two objectives. Re-training LLM2CLIP from scratch on datasets like Laion2B (Schuhmann et al., 2022) and Recaption-1B (Li et al., 2024b) is also a promising direction that we did not pursue due to time constraints.

## 6 CONCLUSION

This paper presents a method that enables LLMs to assist in CLIP training. Given the powerful text understanding capabilities of LLMs, we aim to harness their potential to enhance cross-modal representation learning, allowing language — the carrier of human knowledge — to play a pivotal role in spreading the intelligence of LLMs across multiple modalities and applications. The core contributions of this work are threefold:

First, we identified the primary reason why LLMs struggle to directly participate in multimodal representation learning — the lack of discriminability in their output features. Second, we introduced the caption contrastive fine-tuning method, which significantly improves feature discriminability, removing the biggest obstacle to utilizing LLMs in CLIP training. Third, we designed the LLM2CLIP training framework, which has proven to be an efficient and effective approach for significantly enhancing the performance of pretrained CLIP models.

We experimented with LLM2CLIP on both CLIP and EVA models and validated it across different Llama3 models, including 1B, 8B, and 13B, all of which yielded transformative improvements in CLIP performance. LLM2CLIP offers several advantages, such as seamless compatibility with long and complex text inputs, and the ability to incorporate the open-world knowledge of LLMs into CLIP training — as demonstrated by its ability to match Chinese text even when trained solely on English data.

We firmly believe that the potential of LLMs extends far beyond what has been shown, and we hope this work serves as a starting point for using LLMs to enhance various vision foundation models, ultimately spreading their general capabilities and open-world knowledge across a wide range of modalities and applications.

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
