# OpenReview forum: "LLM2CLIP: Extending the Capability Boundaries of CLIP through Large Language Models"
_ICLR.cc/2025/Conference — ICLR 2025 Conference Withdrawn Submission_

### Official Review · Reviewer_cJ1y · 2024-10-16

**Soundness:** 3
**Presentation:** 1
**Contribution:** 2
**Rating:** 3
**Confidence:** 4

**Summary:**

The paper contributes by adapting vision LLMs for encoding tasks by following LLM2Vec and applying CLIP-style training, showing top results on image-text retrieval benchmarks. However, it lacks novelty in some contributions, fails to properly validate its approach against LLM2Vec finetuned models, and has limited evaluation on non-retrieval benchmarks. Claims are overstated and critical implementation details are missing, hindering reproducibility.

**Strengths:**

The paper presents a contribution by following the LLM2vec technique to adapt large language models (LLMs) for encoding tasks. It then applies CLIP-style training with the addition of adapter and projector layers, which achieved strong results across multiple image-text retrieval benchmarks.

**Weaknesses:**

**Technical:**

1. Section 3.1: The poor performance of LLMs in raw retrieval tasks is expected since they are trained primarily for generation. The LLM2Vec method makes them suitable for embedding and discrimination tasks. However, the paper does not explore or compare against LLM2Vec-transformed models, which could have provided a stronger evaluation baseline.

1. Section 3.1: The first of the three major contributions, "NATIVE LLMS ARE INEFFECTIVE TEXT ENCODERS FOR CLIP," lacks novelty. As LLMs are generally known to perform poorly in encoding tasks without further adaptation, this result is not surprising and applies beyond just CLIP.

1. Section 3.1: I don’t undertsand clearly why the text experiments, such as those done on Llama, help infer encoding capability within CLIP. The text-only experiments don’t seem directly relevant to the cross-modal task that CLIP targets.

1. Section 3.1: The paper claims to introduce "a new metric: the MS COCO Caption Retrieve Accuracy (CRA)," but this metric is not actually novel.

1. Section 3.2: The second contribution is essentially a replication of LLM2Vec, tested with different datasets. Much of Section 3.2 could have been covered in the introduction or related work sections, as it does not offer significant new insights.

1. Section 3.1 and 3.2: Both sections 3.1 and 3.2 could have been avoided if the authors used the LLM2Vec encoder, which is already cited in the paper. The reasoning for not adopting this approach remains unclear, and the decision should be validated with ablation studies.

1. Section 3.3: The justification for not training the base LLM is weak. The claim that large batch sizes are required for effective training is not well-supported, especially since similar methods like SigCLIP have demonstrated success without large batch sizes. Additionally, the use of LoRA (Low-Rank Adaptation) could have been applied to fine-tune the LLM without losing the knowledge base much.

1. Line 283: The details of the adapter module are under-specified, hindering reproducibility.

1. Section 4.1: The training hyperparameters are also not fully disclosed, further limiting reproducibility.

1. Section 4.2: **The claim that knowledge from LLMs unlocks "richer visual representations" is primarily tested on image-text retrieval benchmarks. However, the paper lacks evaluations on classification and purely visual representation tasks, which are essential to validate this claim.**

1. Section 4.2: **Evaluation on purely visual understanding tasks is even more critical since the other compared models have much smaller text encoders compared to the one used in LLM2CLIP.**

**Writing and Presentation:**

1. Abstract: It could be more concise with more clarity. It seems generated directly from the main content using LLM.

1. Figure 1: It is not easy to understand due to the use of multiple sub-images, and the caption is poorly aligned with the figure. Breaking this figure into multiple simpler visuals with clear explanations would enhance comprehension.

1. Table 2: the value "9.2" highlighted in the first column seems to be an error, as it is highlighted as the highest, which contradicts the data presented.

**Questions:**

1) Table 1: How are the text embeddings computed for the models before fine-tuning? Is the last token used, or some pooled output of all tokens?

2) Line 113: “Specifically, we designed a caption contractive (CC) fine-tuning strategy, applying lightweight fine-tuning to the output tokens of Llama-3 8B using LoRA on the CC3M.” What does it mean to fine-tune only the output tokens? Could you clarify this process?

---

### Official Review · Reviewer_fXxX · 2024-10-22

**Soundness:** 3
**Presentation:** 1
**Contribution:** 2
**Rating:** 3
**Confidence:** 3

**Summary:**

The paper proposes a recipe to replace the text encoder of CLIP models with a decoder-only large language model (LLM). They argue that using an LLM would improve the textual understanding of CLIP models and allow them to handle long and complex captions.

They compute the LLM embedding by average pooling all tokens and fine-tune these representations with a procedure similar to LLM2Vec by:

1. Enabling bidirectional attention.
2. Fine-tuning the model via masked next token prediction on captions and Wikitext data.
3. Performing contrastive fine-tuning using both captions and paired English documents.

These fine-tuning steps are all performed via LoRA.

Once fine-tuned, they attach "several linear layers" to the LLM and a projection layer to both models and fine-tune them using a CLIP loss. The LLM is frozen while all other parameters are trainable. They claim this fine-tuning procedure has the same compute burden as training the original CLIP model.

They train models using 3, 15, and 60M captions, using a 1:1 mix between original and rewritten dense captions generated by MLLMs. They also train a variant, LLM2CLIP-S, using no dense captions. They use Llama 3.2 1B, 8B, and Mistral-Nemo 12B as the LLM backbones.

They show that contrastive fine-tuning with captions improves their caption-retrieval score (Table 6) compared to fine-tuning using only generic text. Moreover, they show that models trained with dense rewritten captions perform significantly better on tasks whose images have dense descriptions (Tables 4 and 2). This result also applies to encoders such as Jina-BERT.

They also show that the multilingual abilities of LLMs transfers to image-text retrieval, as LLM2CLIP outperforms all other language-specific models in image-text and text-image retrieval on Chinese captions (Table 3), and that LLM2CLIP can improve performance the performance of Llava on some benchmarks.

Finally, they also show performance scales as more training samples are used, and that the ratio of standard vs dense captions from 1:1 to 1:3 is optimal.

**Strengths:**

The paper proposes a novel idea (replacing the CLIP text encoder with an LLM) and illustrates a recipe for doing so. The method is evaluated chiefly on image-text and text-image retrieval tasks, both in English and Chinese, using seven datasets and shows great results, especially in tasks with dense captions. They also show that their method can, in principle, improve other models that rely on CLIP, such as Llava 1.5.

**Weaknesses:**

The recipe proposed by the paper lacks several crucial details:

- What is the design of the LLM adaptor? The paper only states that it's composed of "several linear layers". Is it an MLP? How deep is it? What activation function is used?
    - Moreover, the fine-tuning hyperparameters are not specified. What optimizer, learning rate (and schedule), etc., was used?
- The LoRA-based fine-tuning uses the same parameters as LLM2Vec, but that paper does not experiment with, e.g., Mistral Nemo. Were the training parameters for `Mistral-7B` used in that case? Overall, I think it would be good to report the training parameters fully for reproducibility.

In addition to the lacking details, the claim that the Chinese language experiment is OOD because they only trained models using English data does not really hold, as Llama 3 was pretrained with multilingual data.

Presentation-wise, the paper was somewhat hard to read and in a state closer to a draft than to a good submission. The paper also uses some expressions that are more appropriate for a product release rather than a scientific article, e.g., "like a miracle", "super text encoder", "audacious step", etc. Some typos and presentation improvements are listed below:

1. (Table 2) In the CLIP + LLM2CLIP / Flickr30k (I2T), the performance is only 9.2, which I assume is a typo as it's bolded.
2. (Table 2) The bolding in the ShareGPT4V (T2I) column and the first block are incorrect. The highest value is on the bottom row, while the penultimate value is highlighted.
3. On line 473, there is a missing table reference (Table X).
4. Bolding issue in Table 7 / Seed Benchmarks (Video).
5. The numbers on line 387 (51.7/50.6) do not match what was reported in Table 4 (32.3/39.3).
6. On line 178, the authors refer to "T5-V", while the method in the cited paper is named "E5-V"
7. The references should be updated, e.g. imu2clip is an EMNLP findings 2023 paper, but its arXiv version is being cited
8. On line 477, it's Mistral NeMo, not "Namo"
9. Line 104, MS-COCO is not cited correctly
10. Line 232, "replacing CLIP's text encoder and the corresponding ViT with Llama-3 8B". This sentence is not clear
11. The metrics are unclear, e.g., what does "@X" mean in Table 3? Recall@X?

**Questions:**

1. The paper states that "the additional training parameters are roughly equivalent to those in the original CLIP, minimizing the overhead". Does this mean that the adaptor has the same number of parameters as the original text encoder (i.e. 130M)? If yes, why is this needed? Would a smaller adaptor work equally well?
2. On line 302 the authors claim that the training overhead is nearly identical to fine-tuning the original CLIP model, does this include the LLM inference step?
3. On line 180, the authors claim that ViTs inherently lack complex visual feature extraction capabilities. No work supporting that claim is cited, and to my understanding, ViTs can have strong emerging properties (e.g., DINO). What did the authors mean exactly?
4. It would be interesting to discuss other methods to extract embeddings from decoder-only LLMs, such as [1, 2] in the related works.

[1]: Fine-Tuning LLaMA for Multi-Stage Text Retrieval, SIGIR 2024

[2]: Improving Text Embeddings with Large Language Models, ACL 2024

---

### Official Review · Reviewer_oroc · 2024-11-01

**Soundness:** 2
**Presentation:** 1
**Contribution:** 2
**Rating:** 3
**Confidence:** 4

**Summary:**

The paper introduces a CLIP-style model that relies on an LLM for the text-encoder to improve natural language understanding for both image and text retrieval tasks. They take a pre-trained LLM, fine-tune it on caption data using a masked next token prediction and then use a contrastive loss (treating matching captions as positive pairs, and all other captions as negatives). Usual CLIP-training follows, initializing the vision encoder with CLIP’s vision encoder weights, and using the new text-encoder with an adapter. Text encoder is frozen, and the usual project layers are added on top of both text and vision encoders. By only training the text adapter, text projector, vision encoder and vision projector in the final CLIP-style training, they significantly improve retrieval performance. Both image to text, and text to image retrieval is improved. Next to that, they show that by relying on the pre-trained LLM, they can do good retrieval on datasets that were originally out of distribution for the vision encoder (shown for Chinese image/text retrieval).

**Strengths:**

1. Relying on the pre-trained LLM allows them to do retrieval on datasets that are out of distribution for the vision encoder, which they show for Chinese image/text retrieval. This shows it can outperform existing models that were trained specifically for Chinese image/text retrieval.

2. They compare various model sizes of the same Llama family, showing the impact of model size.

3. All trainings were done using publicly available data and models.

**Weaknesses:**

1. Big claims, that feel not supported enough.

    1. “To validate the knowledge transfer capabilities of the LLM, we designed an out-of-distribution (OOD) experiment by conducting an image-text retrieval task in a completely unfamiliar language” (page 9) \-- Add details on whether the LLM was trained on Chinese data.

    2. Too many marketing words that distract from the science.

        1. You refer to the encoder as “super text encoder” (page 5) in section 3.3, and section 3.3 only.

        2. “transformative leap in performance” (page 10)

        3. “the magical power of LLM2CLIP became evident” (page 9) \-- It’s not magic, it’s science.

2. Writing/presentation is poor.

    1. Same sentences are repeated 2-3 times (e.g. “CLIP is one of the most important multimodal foundational models today” (page 1)).

    2. Sections contain information from other sections.

    3. References are not in order of mention in the text

    4. Add more details to all tables. What models are shown (what LLM size)? Some tables show big improvements, but they don’t indicate the increase in number of parameters for the text encoder versus the baselines.

    5. For Table4, “Table 4: Ablation Study of LLM2CLIP. Here LLM2CLIP-S refers to the results trained on the original short caption dataset.” (page 8), specify what the “Ablation Study” is. What was tested/compared here?

    6. (nit) Abstract feels very long, and does not seem to be a summary.

    7. “51.7/50.6” (page 7, mentioned in section 4.2) These numbers don't exist in that table.

    8. Typos:

        1. “Mistral Namo” (page 9) \-> Mistral-Nemo

        2. “In Table X, we” (page 9) -- forgot reference

3. Missing references to existing works.

    1. “switching to bidirectional attention improves its ability to capture contextual information” (page 5) \-- cite a work that proofs this, or show it?

4. Missing motivation versus existing works.

    1. Why use a next-token-prediction model, and then train it using Masked Language Modeling (MLM), instead of using an MLM as initialization? Larger Bert-style models are not compared with, but they are also not mentioned as an alternative (or if you don’t think they are, why?).

    2. Why not compare with existing text embedding models that are used for RAG? Given the cheap claimed final CLIP-training, and freezing the language model, taking an off-the-shelf embedding model from HuggingFace would be a great baseline.

5. Missing ablations for choices.

    1. Do we really need MLM, or would an adapter (with next-token prediction) be enough to get better embeddings if we finetune on the caption dataset? Is the reason we see low performance on caption retrieval because the embeddings are generally bad (current claim) or because the captions are not aligned well with the original training data?

6. Missing details

    1. Given you’re using open-source models for initialization of all weights, and the training is ‘light-weight’, can you share the training hyper parameters/recipe? Learning rate? Weight decay? That would allow others to reproduce it.

**Questions:**

1. Our findings indicate that the best performance is achieved when dense captions make up 50% to 75% of the training data, striking a balance between both short-text and long-text retrieval tasks” (page 10)

    1. So why do you use 50%? 75% seems to perform better, right?

2. “However, unlike BERT, we adapt this process to fit the nature of LLMs by predicting tokens just before the masked token” (page 5)

    1. Can you elaborate? What tokens are used in the loss function? From this sentence, I concluded that the loss is computed using the non-masked token before the masked token, but how is the masked token used? Do you use the masked token as target at any point?

3. I might have missed it, since many tables do not contain descriptions, but is there any ablation of the two-stage text-encoder training? I do see caption finetuning results (using the masked next token prediction), but I don’t see any ablation showing the impact of the contrastive text-only stage (described in last part of 3.2).

4. “We then perform caption-to-caption retrieval and assess Top-1 accuracy using different language models, defining the result as their CRA score” (page 4)

    1. How is the retrieval done? From text later, I guess in embedding space? How do you compare embedding distance?

5. “we pre-extract all the text features from the training data and store them in memory” (page 6)

    1. You store all of them in memory? Is that GPU memory or RAM? How much is required for this? Running the inference offline and saving to disk would have given the same LLM inference computation reduction, right?

---

### Official Review · Reviewer_9RC1 · 2024-11-04

**Soundness:** 2
**Presentation:** 2
**Contribution:** 1
**Rating:** 3
**Confidence:** 5

**Summary:**

The paper argues that CLIP's effectiveness has been hindered by its text encoder as more powerful alternatives for language modeling have been proposed, i.e. LLMs, trained with longer contexts and larger text corpora. The paper presents empirical results for how LLM-based representation of image captions are not very discriminative, and thus direct use of these models is not sufficient to address weaknesses in the CLIP model. The paper proposes a strategy that involves tuning the LLM-model (LLama3-8B) with image captions using a contrastive framework so that it learns to discriminate between captions corresponding to the same image and captions that do not. Then a CLIP model is tuned with this new LLM-tuned model as the text encoder.

**Strengths:**

* The paper convincingly presents the case that CLIP models have limitations in the way they encode text and lacks the generality that LLMs have from vast amounts of text and supervised finetuning.
* The paper presents some insights about the latent space of CLIP and LLM models in representing text that could lead to innovations such as the lack of discriminatibility of image captions for standard off-the-shelf LLMs.
* There is some potential for cross-lingual transfer as presented in one of the experiments at the end of the paper.

**Weaknesses:**

* The main weakness is that the main claim in the paper is not convincingly demonstrated. That the resulting LLM2CLIP model has conclusively take advantage of an LLM replacement for its text encoder without losing the original capabilities of CLIP. The closest result that could be used to demonstrate this would be only in Table 7 as it is the only set of results that test the proposed model on downstream tasks e.g. VQAv2, GQA, MMBench, etc. However, here the results are mixed at best, the proposed method only surpasses the baseline model in some cases, and the results are likely not significant e.g. 79.04 vs 79.68 on VQAv2. I am not insisting on state of the art results here since LLaVA1.5 is not the state of the art for most of these benchmarks but improvement from the proposed LLM2CLIP model is really marginal here.
* A key missing experiment is Imagenet top-1 zero-shot accuracy. One of the biggest challenges of replacing the CLIP text encoders is to do so without affecting zero-shot top-1 accuracy on Imagenet, as this is one of the tasks where CLIP excels the most, as it has a large pre-training on 400M image-text pairs. I would also argue that top-1 accuracy on Imagenet and recognition of large number of categories more generally is the biggest reason for adoption of CLIP as a vision encoder.
* I also consider the paper has some presentation issues although I rank this as less important. e.g. page 9 says in Table X. Also the tables are not in the order they are discussed in the paper and Table 7 which has the most important results in the paper uses a really tiny font. The organization of the paper seems to be in the order of how discoveries were made along the way instead of explaining more concisely how the proposed method works and perhaps provide justifications along the way, or later. It makes it really hard to understand what is being proposed until later in the paper or what are the key results. I also take some issue with wordings such as miracle and magical power in the last paragraph of page 9 as it reads a bit as overclaiming, especially the claim that Chinese is a completely unfamiliar language, as Llama clearly used other languages in training but a casual reader might miss this fact and be mislead by these statements. Instead I would encourage the authors to explain how is this being accomplished rather than just attribute it to magic.

**Questions:**

* CLIP ViT/L-14 achieves 75.3 top-1 zero-shot (no training) accuracy in the Imagenet dataset as reported in the original paper. What is the performance for LLM2CLIP?

---

### Note · Authors · 2024-11-12

**Comment:**

We weren’t fully prepared, so we really appreciate your review feedback.

Our new version has shown significant improvements, and we’ll be refining our writing as well. We hope this work will make a truly valuable contribution to the community.

**Withdrawal Confirmation:**

I have read and agree with the venue's withdrawal policy on behalf of myself and my co-authors.